# Design and Performance Evaluation of a Novel Slave System for Endovascular Tele-Surgery

**Chaochao Shi [1], Shuxiang Guo [1,2,*] and Masahiko Kawanishi [3]**

1   Graduate School of Engineering, Kagawa University, Takamatsu 761-0396, Japan
2   Key Laboratory of Convergence Medical Engineering System and Healthcare Technology, Ministry of Industry and Information Technology, School of Life Science and Technology, Beijing Institute of Technology, Beijing 100081, China
3   Faculty of Medicine, Kagawa University, Takamatsu 761-0793, Japan
*   Correspondence: guo.shuxiang@kagawa-u.ac.jp

**Abstract:** Vascular interventional robots have attracted growing attention in recent years. However, current vascular interventional robot systems generally lack force feedback and cannot quickly clamp the catheter/guidewire. The structure of slave systems is unstable and the power transmission is imprecise, increasing the system's safety hazards. Vascular intervention robots generally do not follow traditional surgeons' operation habits and, thus, it is not easy for them to understand and learn how to operate. Therefore, a novel vascular intervention system is proposed. The slave system can quickly clamp the catheter/guidewire, is compatible with various standard catheter/guidewire sizes, has precise power transmission, and has a stable structure. The surface of the catheter/guidewire is clamped without damage. Whether it is on the master side or the slave side, it follows the habits of traditional operators to a great extent. The results show that the measurement accuracy of the axial force meets the requirements of robot-assisted surgery and the system can track the designed position of the catheter/guidewire in real time. This study makes a certain contribution to the development of master–slave systems for endovascular tele-surgery.

**Keywords:** vascular interventional robots; force feedback; catheter; slave system; power transmission





## 1. Introduction

According to current World Health Organization data, cardiovascular and cerebrovascular diseases are the most common causes of human death [1]. With the development of medical technology, vascular interventional surgery has become the primary practice for diagnosing and treating vascular diseases, such as angiography, thrombosis, and vascular sclerosis [2–5]. Doctors usually manipulate the medical catheter/guidewire into the target position in a patient's blood vessel under the assistance of a digital subtraction angiography (DSA) system during the vascular interventional surgery.

Doctors need to wear lead suits weighing up to 20 kg in conventional vascular intervention procedures to avoid long-term X-ray exposure. However, the doctor's hands, face, neck, and eyes remained exposed, making them prone to cancerous tumors and eye disease over time [6–8]. In addition, doctors who wear heavy protective suits for extended periods can experience neck and back pain and injuries [9]. Robot-assisted systems can effectively mitigate these occupational risks. In an interventional study, robot-assisted systems can effectively reduce the median radiation dose to operators by 95.2% [10]. A recent study showed that robot-assisted systems also significantly reduced the radiation dose to the patient [11]. Furthermore, robot-assisted systems may avoid contact between the operator and the patient through remote operation, reducing the problem of infection caused by contact with the patient, such as COVID-19 [12]. Based on these advantages, there is potential to improve surgical safety by studying and using endovascular intervention robotic systems in realistic clinical environments [13].

## 1.1. Related Work

Many of the studies focusing on robot-assisted systems mainly involve commercial companies and university institutions.

At present, commercial robotics, such as Niobe (Stereotaxis Inc., St. Louis, MO, USA) [14], Sensi (Hansen Medical Inc., Mountain View, CA, USA) [15,16], Amigo (Catheter Precision Inc., Ledgewood, NJ, USA) [17,18], and CorPath (Corindus Vascular Robotics., Waltham, MA, USA) [19] are in rapid development (more detailed information is shown in Table 1 [20]). However, the above systems have not gained widespread popularity, mainly due to their complexity and high cost.

**Table 1.** Commercial robotic systems.

| System (Company) | Technology | Clinical Needs | Features | Main Shortcomings | Visual Feedback | Force Feedback |
|---|---|---|---|---|---|---|
| Niobe (Stereotaxis) | Magnetic | RF ablation | Dedicated large magnets; dedicated magnetic catheter with a soft tip | Requires specially designed catheter and a room dedicated to the magnets; complexity of the overall system set-up | Yes | No |
| Sensi/Magellan (Hansen Medical) | Electromechanical | RF ablation/vascular procedures | Remote catheter control with dedicated steerable sheaths | Require specially designed sheaths | Yes | No |
| Amigo (Catheter Precision) | Electromechanical | RF ablation | Remote manipulation of standard tip steerable EP catheters | Encumbrance of the system: larger size (101 × 137 × 112 cm); heavier weight (32 kg) [21] | Yes | No |
| CorPath (Corindus) | Electromechanical | Percutaneous coronary interventions | Remote manipulation of the standard guide catheter, guidewire, and balloon/stent catheters | Requires a dedicated single-use cassette to maneuver the interventional devices; using joysticks to control the motion of the intervention devices is not intuitive | Yes | No |

Therefore, we believe that once the above questions can be solved with the combined application of visual feedback and force feedback, these systems' availability and safety will be improved in the future [13].

Currently, many university institutions around the world are researching vascular interventional robots. Wang et al., at Yanshan University [22], introduced a novel type of master–slave system that has good tracking performance and strong robustness. However, its slave manipulator still does not have the reciprocating operation function of the catheter/guidewire that imitates the traditional doctor operation. Song et al., at Hanyang University [23], proposed a robotic VI system. Its slave manipulator has seven degrees of freedom to control the motions of the catheter, guidewire, microcatheter, and micro guidewire. Its master system adopts two master manipulator modes and has been tested. The test results show that the master manipulator with two degrees of freedom has a good effect. However, its master system lacks force feedback. Yang et al., at Imperial College London [24], proposed a pneumatically actuated MR-safe teleoperation platform. It can manipulate endovascular instrumentation and provide operators with force feedback for endovascular tasks. However, the MR-safe Slave Robot cannot manipulate the advancement and rotation of the catheter simultaneously. Yang et al., at Beijing Institute of Technology [25], designed a novel type of master manipulator to imitate the operating habits of

human hands. However, the master manipulator has no torque feedback. The device that generates force feedback uses a motor prone to a LOCKED-rotor state and generates heat, thereby affecting the accuracy of the force feedback. Choi et al., at the University of Ulsan College of Medicine [26], presented a novel robotic system using modularized bi-motional roller cartridge assemblies for robotic vascular interventions. However, it uses rollers to drive the catheter/guidewire, which increases the potential to pollute/damage the surface of the catheter/guidewire. Therefore, the possibility of thrombosis will be increased in actual clinical practice.

In particular, for the slave system, the realization of the catheter/guidewire's motion (advancement/retreatment and rotation) poses new challenges to the structure and the motion transmission form. Bao et al. [27] and Jin et al. [28] proposed that the catheter and guidewire could be co-operated on the slave side. However, the bottom platform of the slave manipulator is only based on a slide rail. During the actual operation, it will sway from side to side, which increases the safety hazard in the system. From the perspective of power transmission of the motor, Bao et al. and Wang et al. [22] used the timing belt or rope transmission for the advancement/retreatment or rotation of the catheter/guidewire on the slave side. It will cause certain deformations and motion hysteresis during the power transmission process because of the flexible characteristics of the timing belt and rope material. Therefore, this type of power transmission increases the instability of the robotic system. In addition, in the design of the clamping mechanism, it is not easy to assemble and disassemble the catheter and guidewire quickly. Its channels are closed. In the process of installing/replacing the catheter and guidewire, it needs to be threaded through a closed channel. This increases the time surgeons spend installing and replacing catheters and guidewires. It also places an additional burden on surgeons and wastes valuable treatment time for patients.

*1.2. Our Contribution*

In this study, a master–slave vascular intervention system is developed and a novel slave system is designed and manufactured based on our earlier research [29]. The slave system has the following characteristics: (1) axial force and radial force measurement of the catheter/guidewire; (2) a stable structure and reciprocating manipulation; (3) the ability to quickly clamp/install/replace the catheter/guidewire; (4) compatibility with various conventional types of the catheter/guidewire; (5) the ability to simultaneously clamp and rotate the catheter/guidewire; (6) the capacity to clamp the catheter/guidewire without damage; (7) an operation mechanism that imitates the operation habits of traditional surgeons.

The remainder of this paper is organized as follows. In Section 2, the overall master–slave system is introduced. The detailed design of the slave system is presented in Section 3. In Section 4, we describe the experimental design and provide the experimental results, which show the performance of the proposed slave system. In Section 5, the discussions are provided. Finally, the conclusions are presented in Section 6.

## 2. Our Robotic System

The overall robotic system is shown in Figure 1. It mainly includes a master side and a slave side. Our robotic system adopts a master–slave teleoperation configuration that allows the surgeon to avoid X-ray exposure. The surgeon operates the operating handling device (simulating the real catheter or guidewire) on the master side to control the slave manipulator to pull/push or rotate the catheter/guidewire into the target position in the patient's blood vessel. During this process, these things will happen simultaneously. From the master to the slave side, the master control unit will record the surgeon's hand information and transfer this information to the slave control unit. The slave control unit will deal with the information to control the slave manipulator to replicate the motion of the surgeon's hand. From the slave side to the master side, the load cell and torque sensor will display the force information of the catheters and guidewires. The slave control unit deals with the force information and transfers it to the master control unit. The master control

unit deals with the force information and controls the master manipulator to provide force feedback to the surgeon's hand. The computer provides visual feedback to the surgeon through the IP camera.

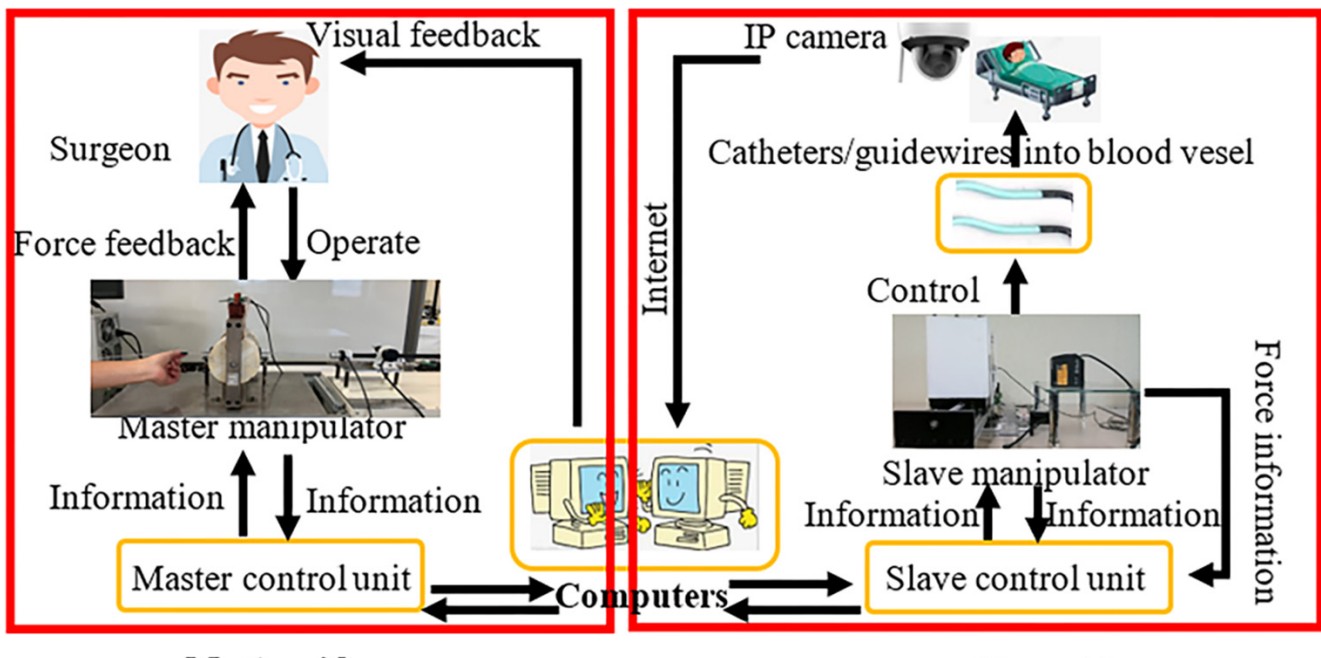

**Figure 1.** Overview of the master–slave system.

### 2.1. Introduction of the Master System

Interventional surgery requires surgeons to manipulate catheters/guidewires into a target position and perform related procedures (such as angiography) in the patient's diseased blood vessels. For human blood vessels, especially brain blood vessels, their volume and shape are small and tortuous. Even a little carelessness can cause the catheter/guidewire to puncture the blood vessel wall, causing patient bleeding and increasing clinical risks. During the vascular intervention surgery, the surgeon must sense the force and the relative position of the catheter/guidewire in real time to prevent mishandling and increase the risk of re-injury to the blood vessel. From this point of view, it is necessary to require the master system to apply haptic feedback to the surgeon's hand and collect the surgeon's hand motion information in real time. In the whole process, in order to improve the understanding ability and learning efficiency of the surgeon with a lack of clinical experience, it is also necessary to require the master system to follow the operating habits of traditional surgeons. This will indirectly increase the safety of the robotic system.

To sum up, the master system needs to have three basic functions:

1.  Recording the radial and axial displacement information of the surgeon's hand when the surgeon is operating the operating handling device;
2.  Providing the surgeon with axial and radial force feedback;
3.  Following the operating habits of traditional surgeons.

Therefore, the following master system was used in this study, as shown in Figure 2a. The master system was designed, manufactured, and developed by doctoral students at Kagawa University [28,30,31]. The first basic function is recorded by two encoders (MTL, MES020-2000p, Japan), as shown in Figure 2b; the second basic function is to provide surgeons with immersive tactile force information by means of the magnetorheological fluid under the action of different magnetic fields. The operating handling device used by the master system is replaced by a rod. It completely follows the operating habits of traditional surgeons when the surgeon operates catheters/guidewires.

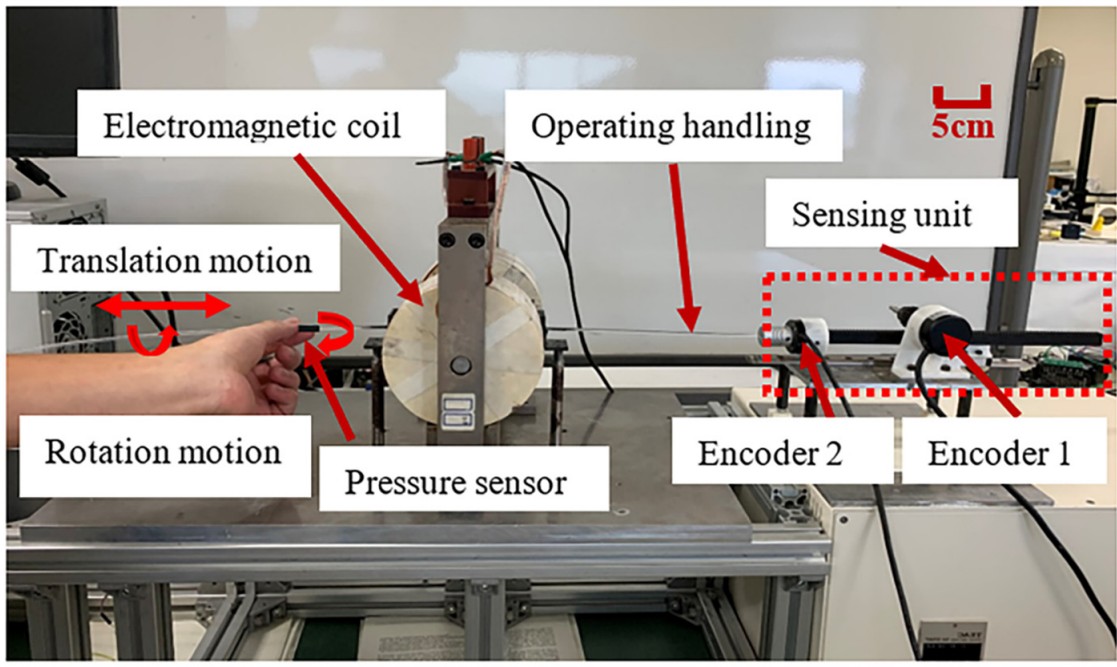

(a)

(b)

**Figure 2.** (**a**) Master manipulator operation interface diagram. (**b**) Operation information recording.

The pressure sensor attached to the operating handling device on the master side can be used to drive the stepping motor on the slave side, realizing the dynamic clamping/releasing of the catheter/guidewire. It is similar to the actual operation of a surgeon's thumb and index finger.

In the master–slave system, the provision of force feedback information lies in the properties of magnetorheological (MR) fluids. Under the action of an external magnetic field, the MR fluids can produce a reversible state-changed response in milliseconds. The particles in the MR fluid are disordered in the absence of a magnetic field. When a magnetic field is applied, the particles become oriented. As the magnetic field strength increases, the particles suspended in the fluid begin to form chains along the magnetic field lines. The apparent viscosity of MR fluids can be changed by adjusting the strength of the magnetic field. Under the action of different magnetic field strengths, the MR fluid will generate a different shearing force on the operating handling device so that it is applied to the surgeon's hand in the form of tactile force. A schematic diagram for the realization form of force feedback is shown in Figure 3 [28,30–32]. The relevant detailed calibration experiments of the master manipulator were reported in our previous study [28]. The

relationship between the force feedback (including axial and radial force feedback) and input current can be obtained.

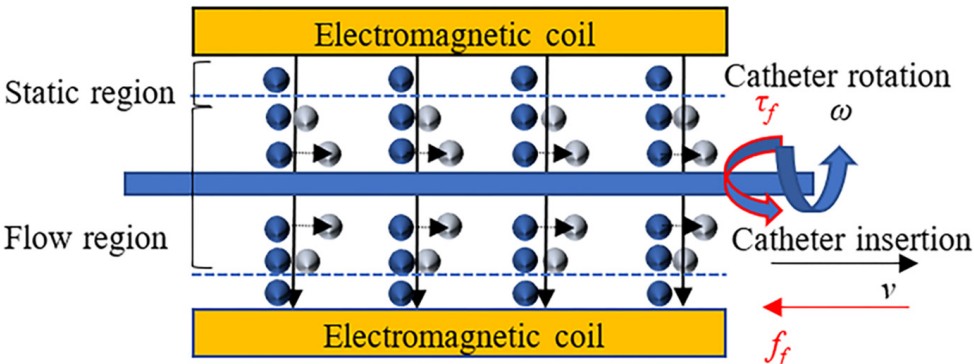

**Figure 3.** Schematic diagram of force feedback generation.

### 2.2. Overview of the Slave System

The role of the slave manipulator is to replace the traditional surgeon's hand to control the motion of the catheter/guidewire. For catheters/guidewires, the slave manipulator needs to realize stable clamping/releasing, advancement/retraction, rotation, and axial and radial force measurement. In addition, the catheter and guidewire can be co-operated on the slave side. The above is the basic function of the slave system. According to the last part of the introduction, with the development of technology, there has been more demand for slave systems.

We proposed the concept of two platforms and added a coordinate system to the system for the first time [29], which increases the stability and accuracy of the system. The use of a silica gel flexible material and clamping device was also discussed.

Based on the above considerations and the basis of previous studies, we designed and proposed a novel slave manipulator, as shown in Figure 4b. The inspiration is shown in Figure 4a. The functions of rotation and clamping/releasing of the catheter/guidewire are realized on the second platform (SP). The advancing/retreating of the catheter/guidewire is arranged on the first platform (FP). The advantage of the design is that the rotation, clamping/releasing, and advancement/retraction of the catheter/guidewire can be manipulated simultaneously.

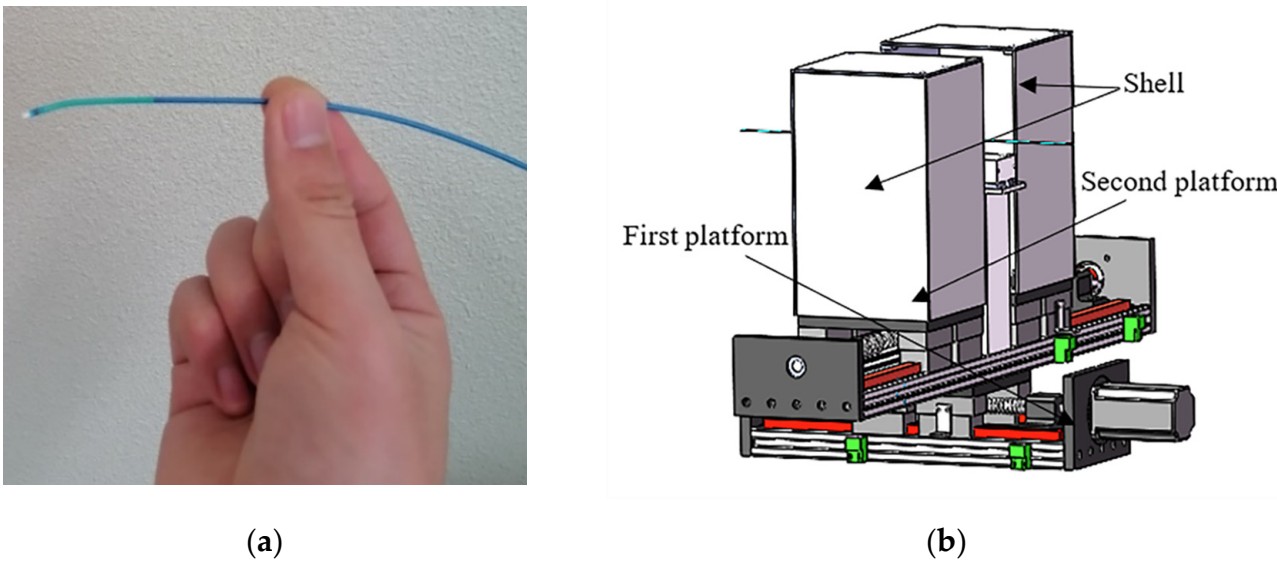

**(a)**          **(b)**

**Figure 4.** (**a**) Traditional manual operation; (**b**) proposed novel slave manipulator.

## 3. Design of the Slave System

### 3.1. Design of the FP

The FP structure is shown in Figure 5.

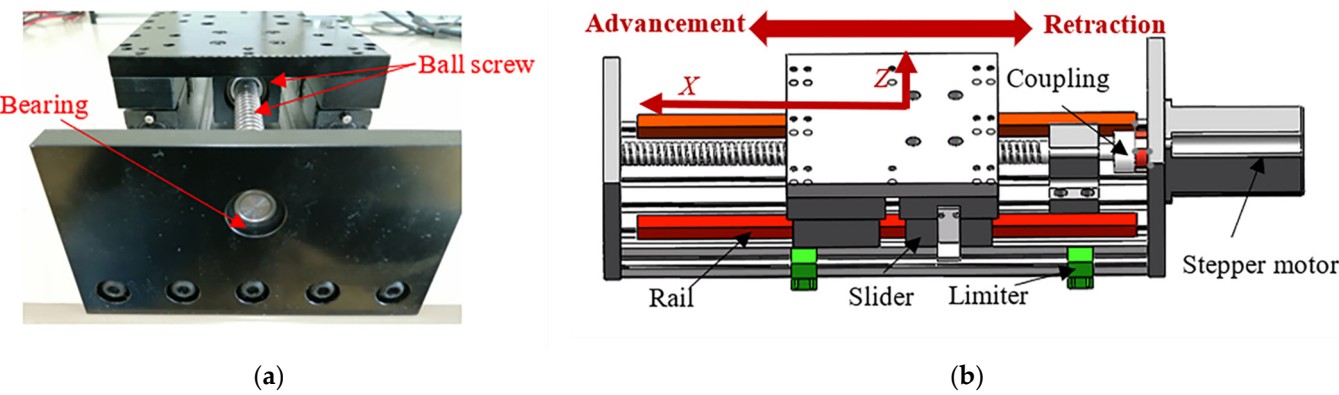

**Figure 5.** (**a**) Left view of the FP (physical picture); (**b**) front view of the FP (designed picture).

As shown in Figure 5a, we chose a combination of a ball screw, rail, and slider in terms of power source transmission. This structure is rigid and the power source transmission is reliable. The ball screw's manufacturing accuracy is greater than that of the timing belt. Considering the structural stability of the slave system and the smoothness of power transmission, we designed a combination of two long sliding rails and four sliding blocks. Through a rectangular steel plate, four sliders, two slide rails, and ball screws are connected to form a stable, rigid connection surface, increasing the system structure's stability and the accuracy of motion transmission.

As shown in Figure 5b, the stepper motor (57CME22, China) is connected to the ball screw through the coupling. The movement of the ball screw drives the movement of the four sliders and the rectangular steel plate. This combination satisfies the advancement and retraction of the catheter/guidewire. Under the forward and reverse direction of the stepper motor (57CME22, China), the rectangular block can move forward and backward along the X-axis. In consideration of the system's safety, two limiters (SN04-N, China) are added to the FP. When the slider moves to the limit of both ends, the system will immediately stop the current movement and provide alarm information, which lays the foundations for the subsequent reciprocating operation.

### 3.2. Design of the SP

As shown in Figure 6a, the ball screw adopts positive and negative threads. Through the movement of the stepping motor, the catheter/guidewire can be clamped/released along the Y-axis synchronously. At this time, the catheter/guidewire only needs to be installed at the center of the slave system. Both sides of the clamping structure have the same design. The rotation of the catheter/guidewire is shown in Figure 6b. Driving two stepping motors (ASM46AA, ORIENTAL MOTOR, Japan) realizes the rotating operation of the catheter/guidewire around the X-axis, similar to the operation of a traditional surgeon's hands. As shown in Figure 6c, it is a rigid power transmission form through adopting the combination of the T-screw and stepper motor to realize the rotation of the catheter/guidewire. Balls are used for transmission between the ball screw, the slider, and the slide rail. Its friction force is almost ignored. Rotation, clamping, and releasing of the catheter/guidewire are realized with rigid structures. These guarantee high precision and timely power transmission in the system. In addition, due to the self-locking property of the ball screw, after the stepper motor is powered off, the current position of the slave manipulator will be maintained without sliding. This indirectly increases the safety of the system. As shown in Figure 7, we tilted the angle $\theta$ to $45°$, or even a larger angle, and the structure on the SP did not slide.

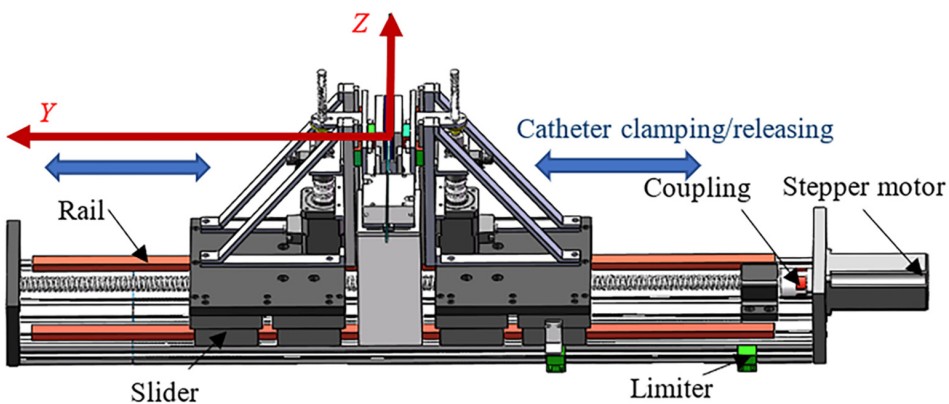

(**a**)

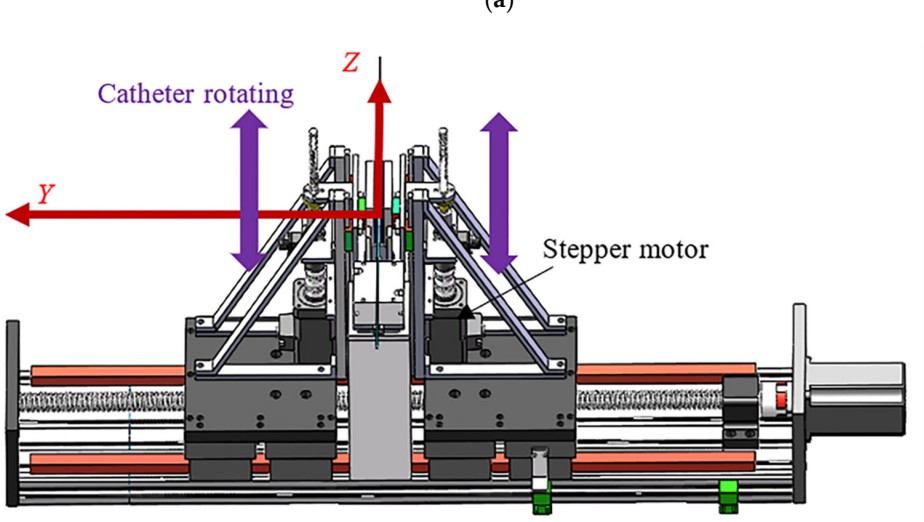

(**b**)

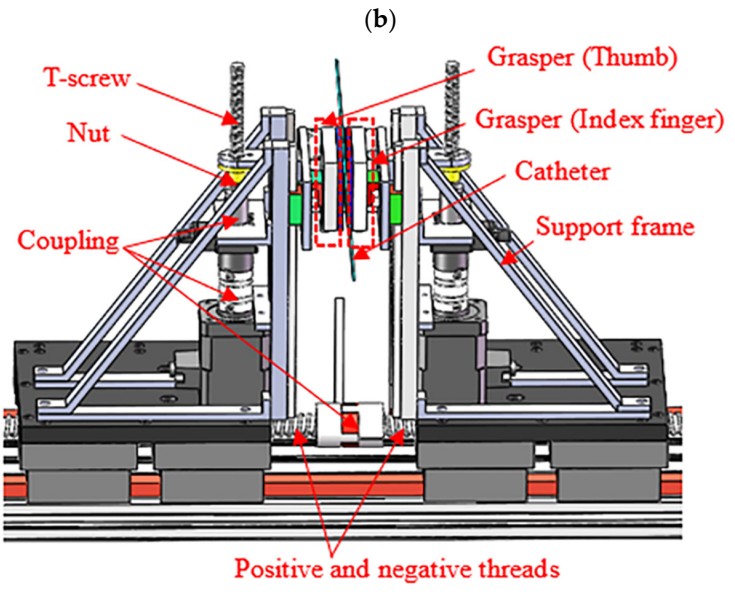

(**c**)

**Figure 6.** Structural design drawing of the SP. (**a**) Clamping mechanism; (**b**) rotation mechanism; (**c**) internal details of the SP.

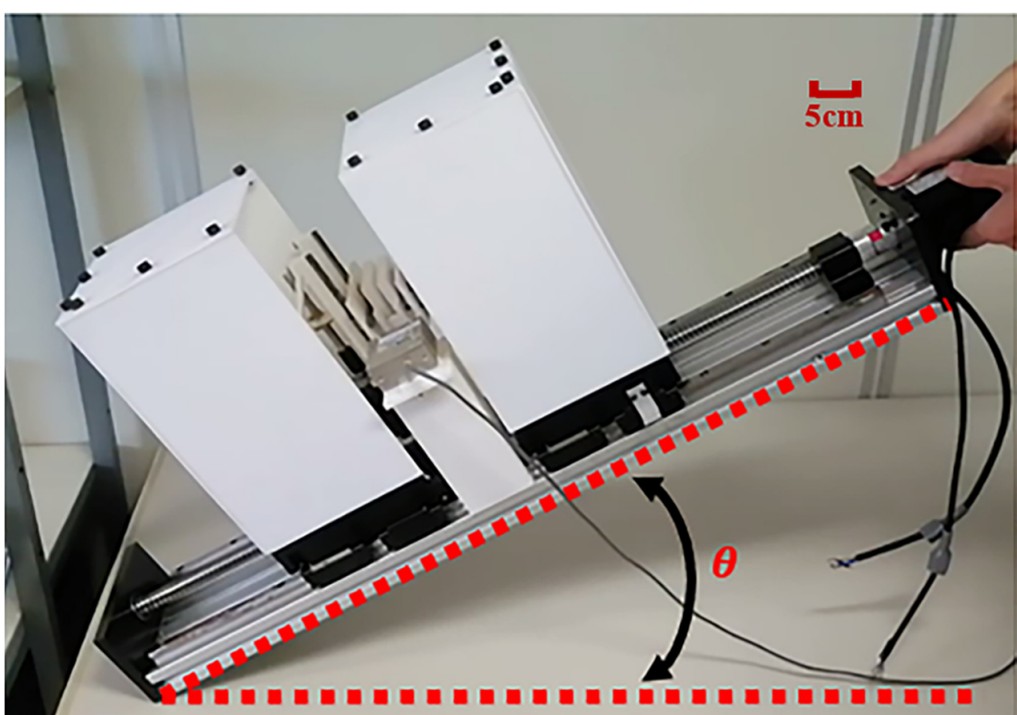

**Figure 7.** Schematic diagram of the SP inclined at a certain angle (*θ*).

### 3.3. Design of the Force Detection Structure

In order to accurately realize the detection of the two forces (axial and radial force), the two-force-detection device is shown in Figure 8a. In this device, the radial force detection of the catheter/guidewire adopts the commercial sensor: the torque sensor (RLW05m, Japan). A coupling connects the torque sensor shaft and the stepping motor shaft (ASM46AA, ORIENTAL MOTOR, Japan). The radial force of the catheter/guidewire is transmitted to the torque sensor through the combination of slider A and the related slide rail. The torque sensor recognizes the torque forces acting on the catheter/guidewire by detecting the slight change in torque in the motor shaft. It can be found that the torque force detection and identification in this study creatively use two torque sensors. Their main function is to compare the same or different torque forces detected by the two torque values. It can be judged whether the catheter/guidewire has slipped during the rotation process, which increases the safety of the operation. In the axial force detection method of the catheter/guidewire, a loadcell (TU-UJ5C, TEAC, Japan) is used to detect the catheter/guidewire's axial force (including insertion force and retraction force). The loadcell standard output is −5N ~ 5N. Its fixing and installation position is shown in Figure 8b. Its fixing and installation position can be fixed and adjusted by manipulating tightening screws. The axial force of the catheter/guidewire is transmitted to the load cell through two graspers (A and B), which are similar to the index finger and thumb of the human hand, and slider B. Grasper A and B are designed with an integrated structure. They have a symmetrical structure to avoid assembly errors, which will affect the transmission of the axial force of the catheter/guidewire. To the best of our knowledge, in the current slave systems with a structural design that can quickly clamp the catheter/guidewire, there is generally a lack of axial and radial force detection of the catheter/guidewire. We are the first to design and manufacture this slave system structure.

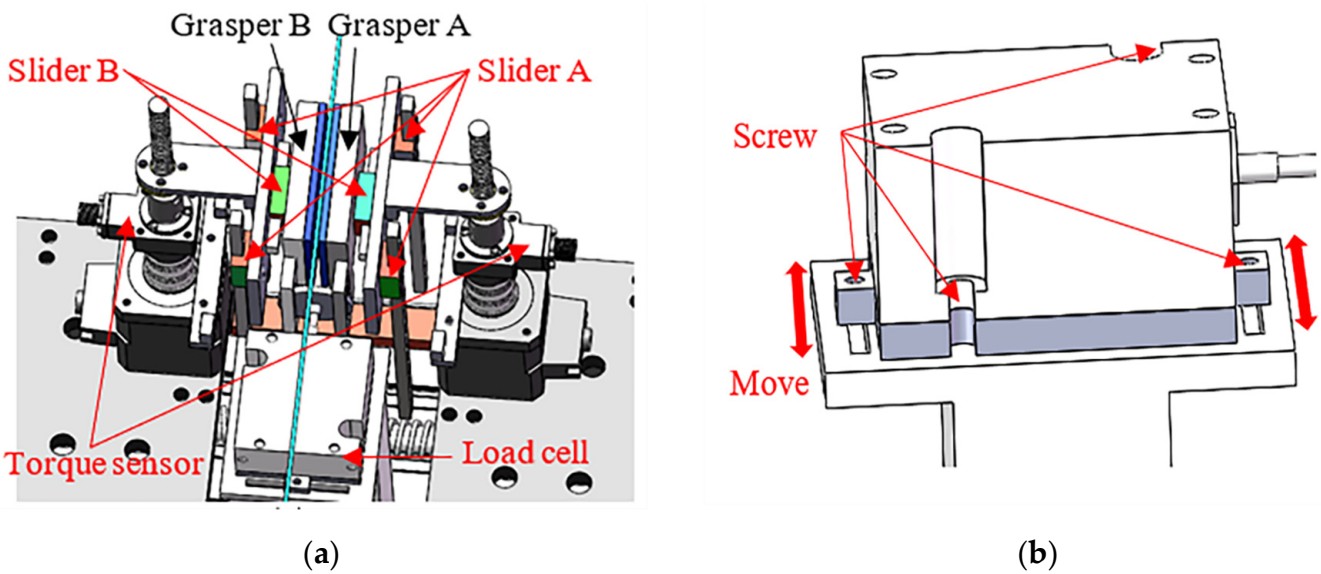

(**a**)　　　　　　　　　　　　　　　　(**b**)

**Figure 8.** Design of the force detection structure. (**a**) Detailed drawing of force measuring device; (**b**) loadcell installation location.

### 3.4. Reciprocating Manipulation

A schematic diagram of the reciprocating manipulation on the slave side is shown in Figure 9. Compared with the use of a roller drive [19,26], this method can effectively reduce the contact area of the surface of the catheter/guidewire and prevent the surface of the catheter/guidewire from being polluted or damaged.

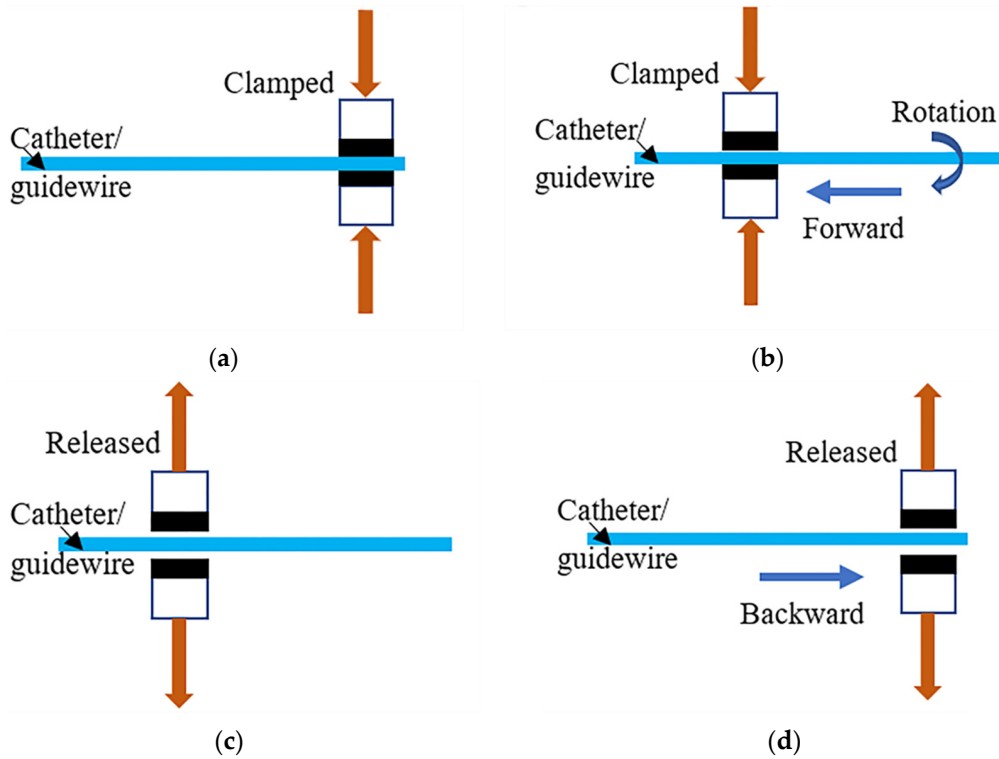

**Figure 9.** A schematic diagram of the reciprocating manipulation. (**a**) At the initial point position, the grasper clamps the catheter/guidewire; (**b**) it clamps the catheter/guidewire for forward or rotational movement; (**c**) after reaching the platform endpoint, it releases the catheter/guidewire; (**d**) the grasper returns to the initial position.

A schematic diagram of the co-operation of the catheter and guidewire is shown in Figure 10. This method can effectively clamp the catheter or guidewire and complete the delivery task together.

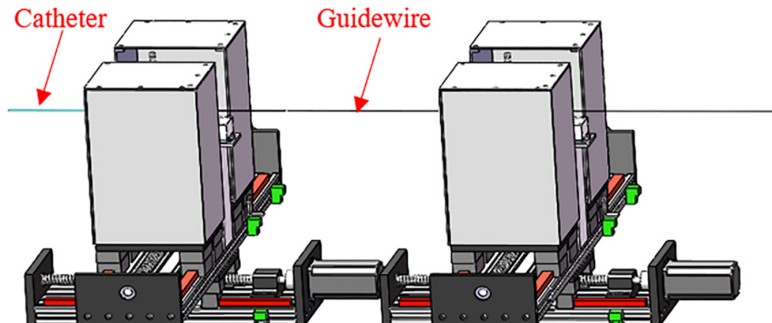

**Figure 10.** Schematic diagram of the co-operation of the catheter and guidewire.

## 4. Experimental Design and Results

### 4.1. Axial Force Calibration Experiment on the Slave Side

We designed the experiment to verify the axial force detection structure's effectiveness, as shown in Figure 11. Here, due to the special treatment on the surface of the catheter/guidewire, we used the rod to replace the catheter/guidewire without affecting the experimental result. The rod material is acrylic. The left side of Figure 11c is a load cell fixed on the ball screw, which is marked as L1 in the paper. The ball screw is connected to the stepper motor (ASM46AA, ORIENTAL MOTOR, Japan) for fixing to prevent the load cell from slipping during the experiment. The load cell shaft is directly connected to the rod through the coupling. The standard axial force is represented by F1. In the slave system, the load cell installation position is on the right of Figure 11c, marked as L2, and F2 represents the detected force. In the experimental design, in order to improve the detection accuracy and avoid experimental errors, the axis of the rod and the shaft of L1 and L2 are on the same horizontal line and the same axis, as shown in Figure 11b,c. The rod is clamped by the SP. When the FP is moving, the results detected by L1 and L2 are shown in Figure 12. The data sampling frequency of L1 and L2 is set to 200 ms.

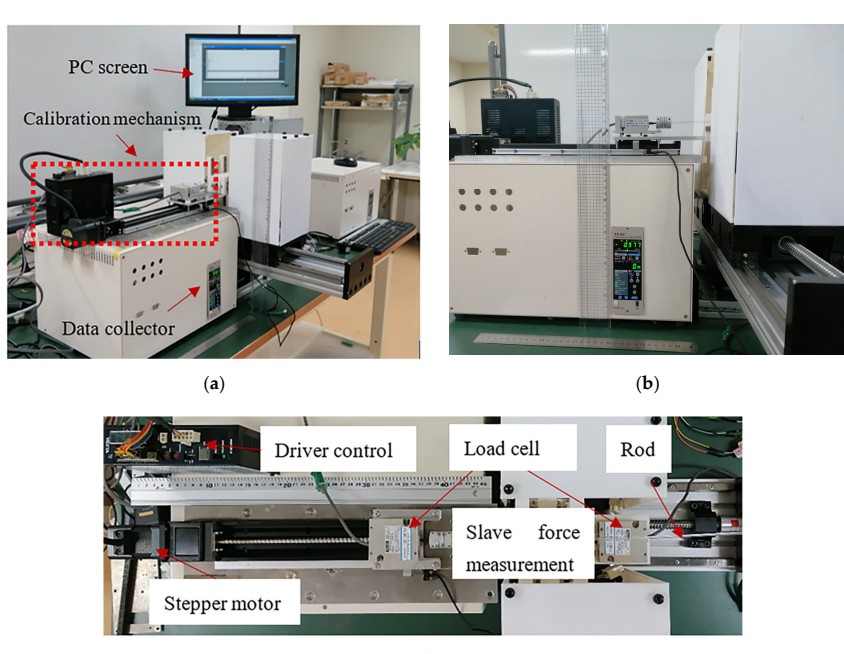

**Figure 11.** Schematic diagram of axial force calibration. (**a**) Full view; (**b**) front view; (**c**) top view.

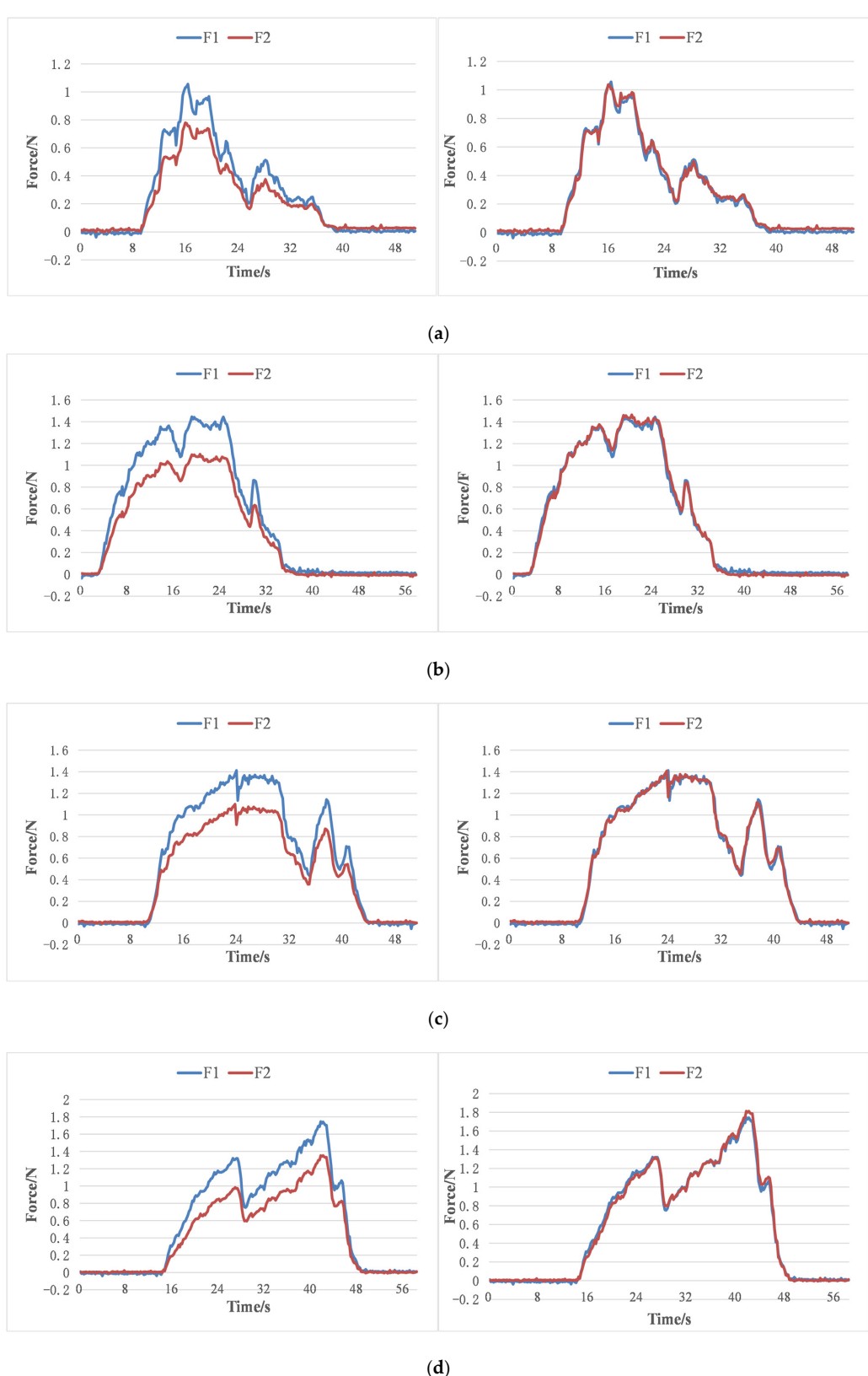

**Figure 12.** Axial force calibration tests were repeated 4 times. (**a**) Original data (**left**) and processed data (**right**) in the first test; (**b**) original data (**left**) and processed data (**right**) in the second test; (**c**) original data (**left**) and processed data (**right**) in the third test; (**d**) original data (**left**) and processed data (**right**) in the fourth test.

As shown in Figure 12, the F1 is greater than F2. L1 and the rod are directly connected through the coupling. Because the detection method is direct, it can be said that F1 is the rod's axial force. The axial force is transmitted to L2 through slider B, the flexible material, and integrated grasper A and B. The force detection method in the slave system is indirect. These materials (the flexible material and 3D-printing materials) are not rigid. They will deform slightly under external force, causing F1 to decrease during transfer to L2. The use of flexible material avoids damage to the catheter/guidewire's surface during clamping. However, it will affect the accuracy of the axial force measurement. It is worth mentioning that due to the influence of the flexible material coefficient, the value of F1/F2 is about 1.33 in this study [33]. F2 × 1.33 is subtracted from F1 to obtain the absolute value of the error and calculate its error percentages, which are 6.31%, 3.84%, 4.93%, and 4.43%, respectively, as shown in Figure 12. According to a previous study, the perceptual resolution of force discrimination (measured by the just-noticeable difference (JND)) is 7–10% in a range of 0.5–200 N [34]. Therefore, we believe that the force detection structural design has sufficient sensitivity to detect small force changes on the catheter/guidewire.

### 4.2. Master–Slave Linear Tracking Test

The advancing and retreating of the catheter/guidewire are realized by connecting the ball screw to the stepper motor (57CME22, China) on the FP. In order to verify the effect of the proposed slave manipulator on the linear movements of the catheter/guidewire, it is necessary to measure the tracking performance of the master–slave system. The experimental setup is shown in Figure 13.

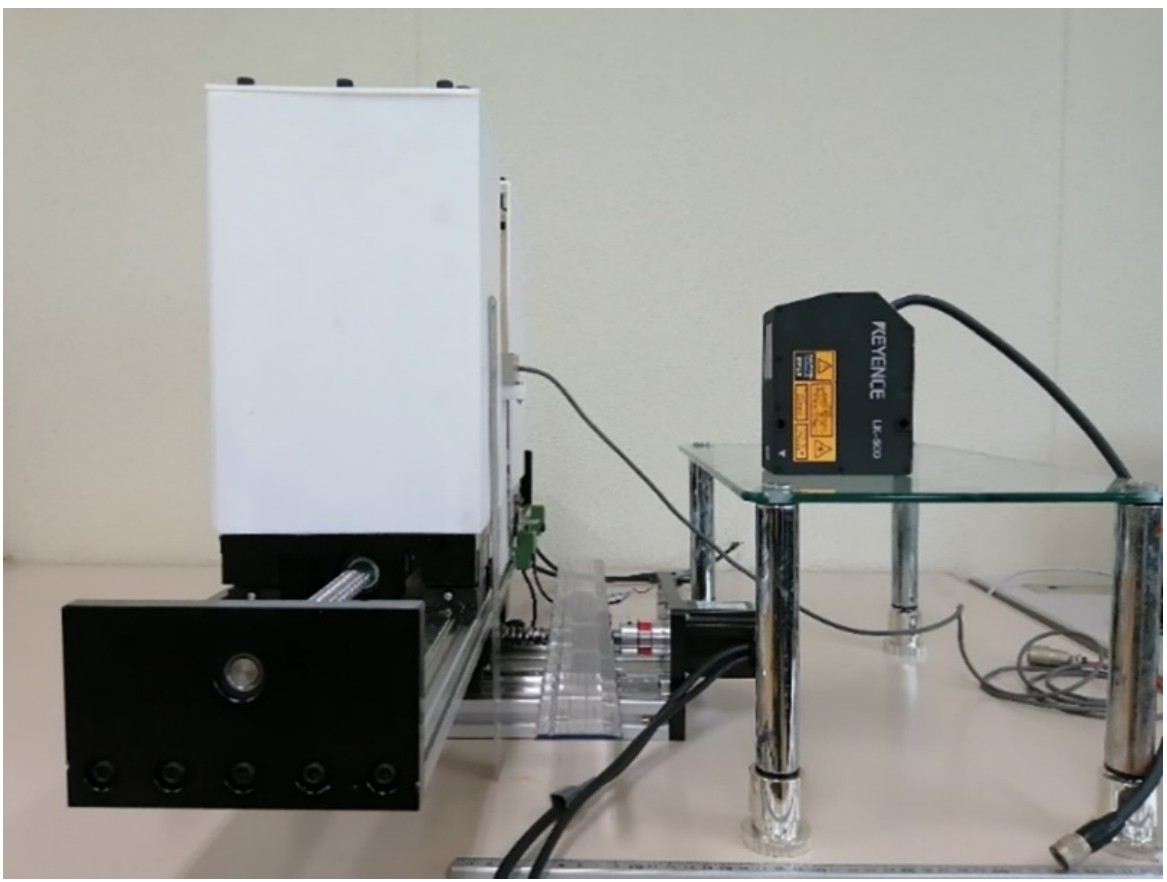

**Figure 13.** Line tracking test setup on the slave side.

As shown in Figure 13, a laser displacement sensor (LK-500, Japan) is employed to detect the displacement of the slave manipulator. The accuracy of the laser displacement sensor is 50 μm/mv, the maximum output power is 15 mW, and the laser wavelength is

690 nm. The ball screw used in the FP is 16 mm in diameter and 5 mm in pitch. A stepper motor driver (CL57, China) drives the closed-loop stepper motor (57CME22, China) to prevent it from losing steps during a sudden movement. The measurement range of the line tracking test is from 0 mm to 200 mm. When the encoder on the master manipulator rotates in a circle, the catheter/guidewire should advance by 10 mm (theoretical value). The control board adopts Arduino mega 2560.

The master–slave line tracking result is shown in Figure 14. As shown in Figure 14, we can find that the greater the speed of the master side, the greater the error that will be produced. When the speed of the master side is about 6.7 mm/s, the average error is about 0.86 mm. When the speed of the master side is about 2.8 mm/s, the average error is about 0.39 mm. When the speed of the master side is greater, the tracking error will be larger. This may be due to a certain time delay.

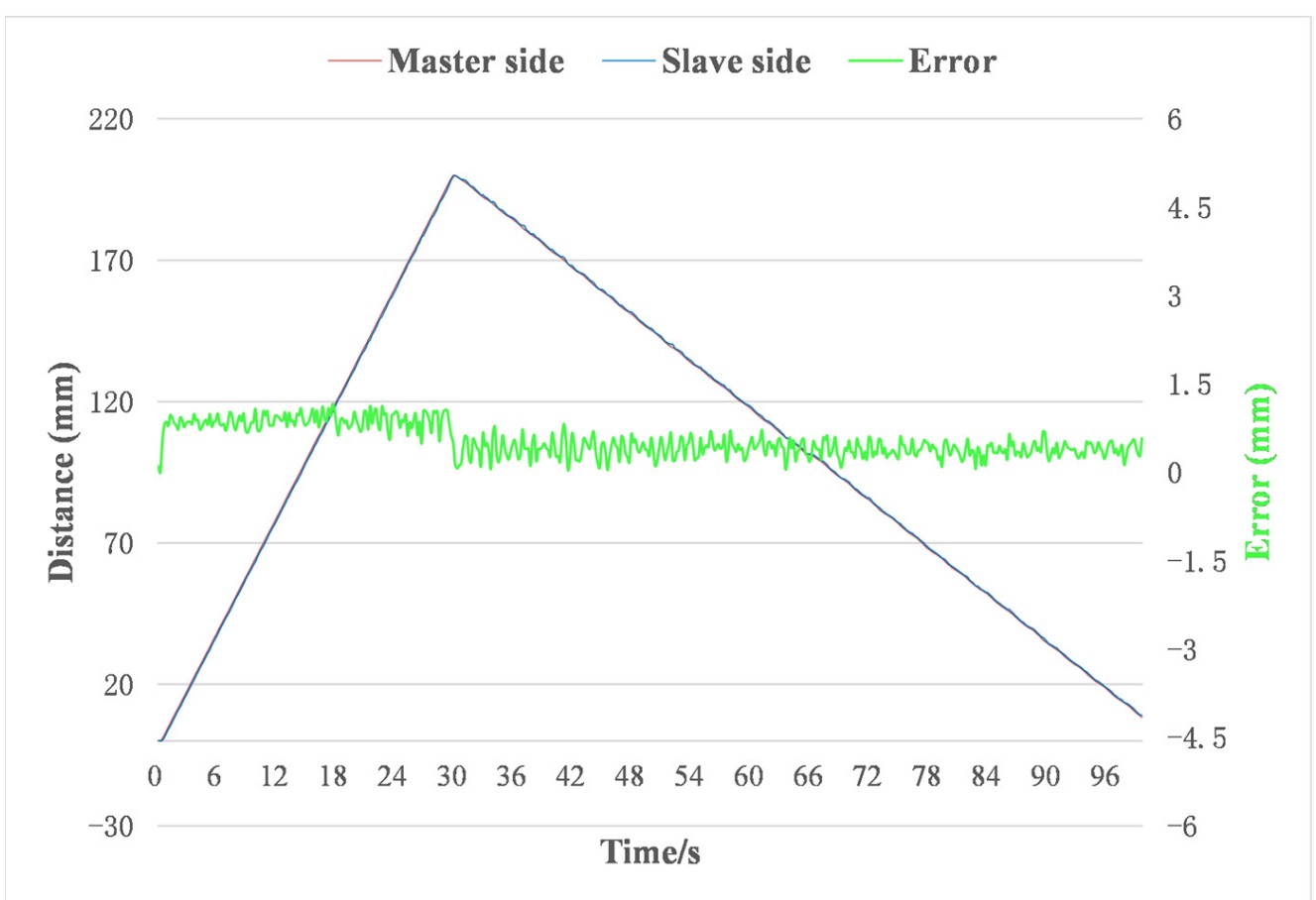

**Figure 14.** Result for the linear tracking in the experiment.

We performed five experiments under the same conditions to check the system's stability. The master–slave line tracking test results are shown in Figure 15 and Table 2. The maximum linear tracking error was 1.36 mm and the average linear tracking error was 0.528 mm. In traditional interventional surgery, experienced physicians operate catheters/guidewires with errors of greater than 2 mm [30]. Therefore, we think that the control system satisfies the requirements of the surgery under this condition. The variance is about 0.07 mm. We can obtain the result that the control system and the slave structure are stable.

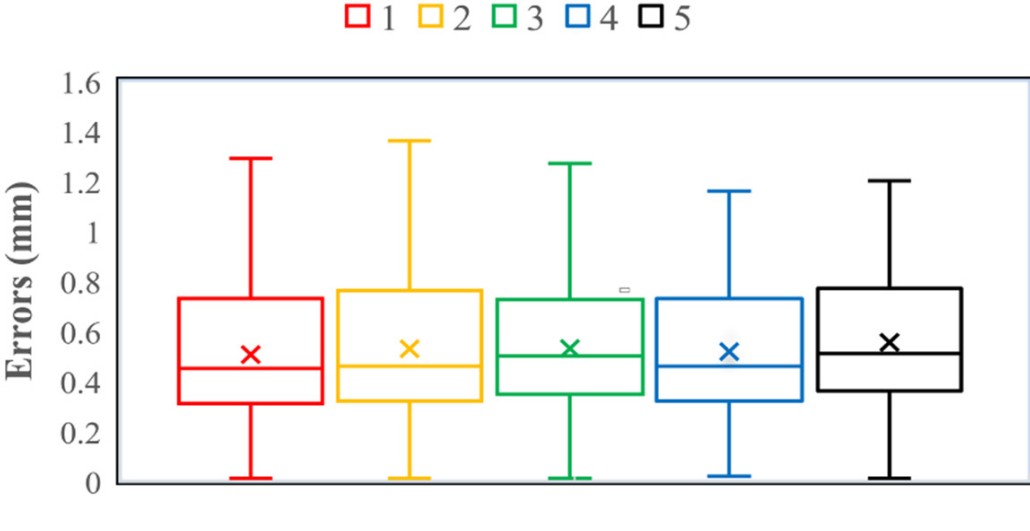

**Figure 15.** The error of linear tracking after the experiment was repeated five times.

**Table 2.** The statistical result of errors in the system's linear tracking.

| Trials | Average (mm) | Variance (mm) | Maximum (mm) | Minimum (mm) |
|--------|--------------|---------------|--------------|--------------|
| 1 | 0.51 | 0.08 | 1.29 | 0.01 |
| 2 | 0.53 | 0.08 | 1.36 | 0.01 |
| 3 | 0.53 | 0.07 | 1.27 | 0.01 |
| 4 | 0.52 | 0.07 | 1.16 | 0.02 |
| 5 | 0.55 | 0.08 | 1.20 | 0.01 |

*4.3. Glass Tube Test*

In order to verify the performance of the force measurement system in practical applications, we performed two experiments using a glass tube with an outer diameter of 7 mm and an inner diameter of 5 mm.

Condition 1: The catheter is inserted alone.

Condition 2: After the guidewire is guided in advance, the catheter is inserted.

In order to study the above two situations, the force of the catheter changes. The experimental setup is shown in Figure 16.

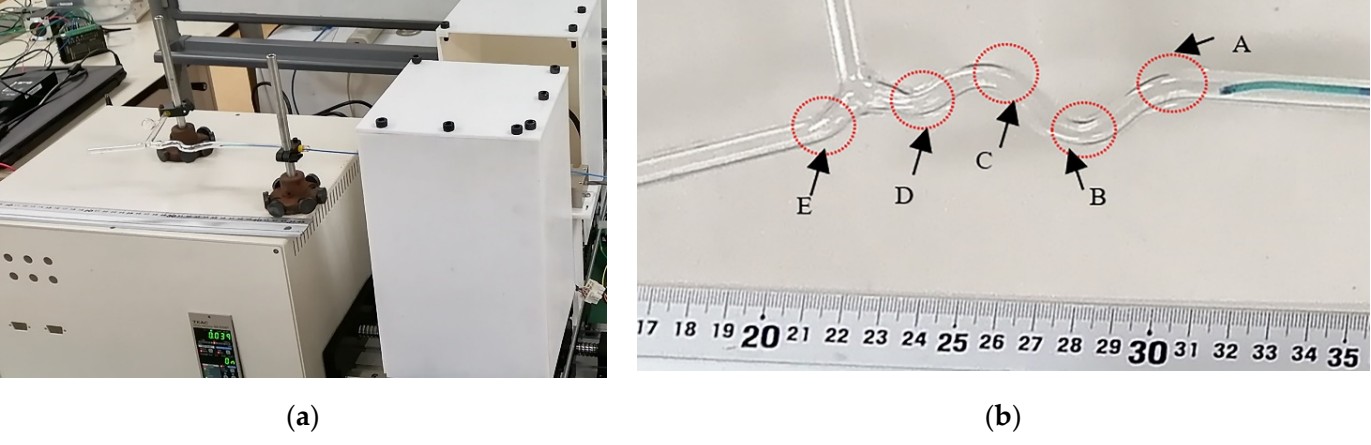

(**a**)           (**b**)

**Figure 16.** (**a**) Glass tube test design; (**b**) inflection points (A, B, C, D, E) in the glass tube.

First, the concept of DCE and CCE was introduced [35]. The discrete contact element (DCE) only touches the glass tube at one or both ends. In contrast, the continuous contact element (CCE) touches the glass tube's walls along its body. In this paper, CCE and DCE refer only to the catheter.

The catheter is a 5 Fr catheter (LA5CAD, USA). The surgeon operates the operating handling device on the master side to control the advancement/retraction of the catheter along the glass tube. When the catheter passes through the glass tube, the axial force of the catheter measured by the slave manipulator is shown in Figure 17 and the details are shown in Figure 18.

As shown in Figures 17 and 18, the force measurement system has a high practical sensitivity. It can reflect the inflection point of the advancement of the catheter in the glass tube. At the same time, we can see that the change point mainly occurs at the position where the tip of the catheter and the glass tube collide. At the '6' point, the force of the catheter is the largest (CCE). At this point, the shape of the catheter is deformed; the entire surface of the catheter is closely attached to the inner wall of the glass tub. In addition, the tip of the catheter and the inner wall of the glass tube are in serious conflict.

The guidewire used in this experiment is a long guidewire with an angle type of 45 degrees. Firstly, the guidewire was inserted into the glass tube in advance manually. Both ends of the guidewire were fixed with tweezers. Then, the slave manipulator controlled the catheter inserted into the glass tube along the guidewire. The test results are shown in Figure 19. The initial situation under condition 2 is shown in Figure 20.

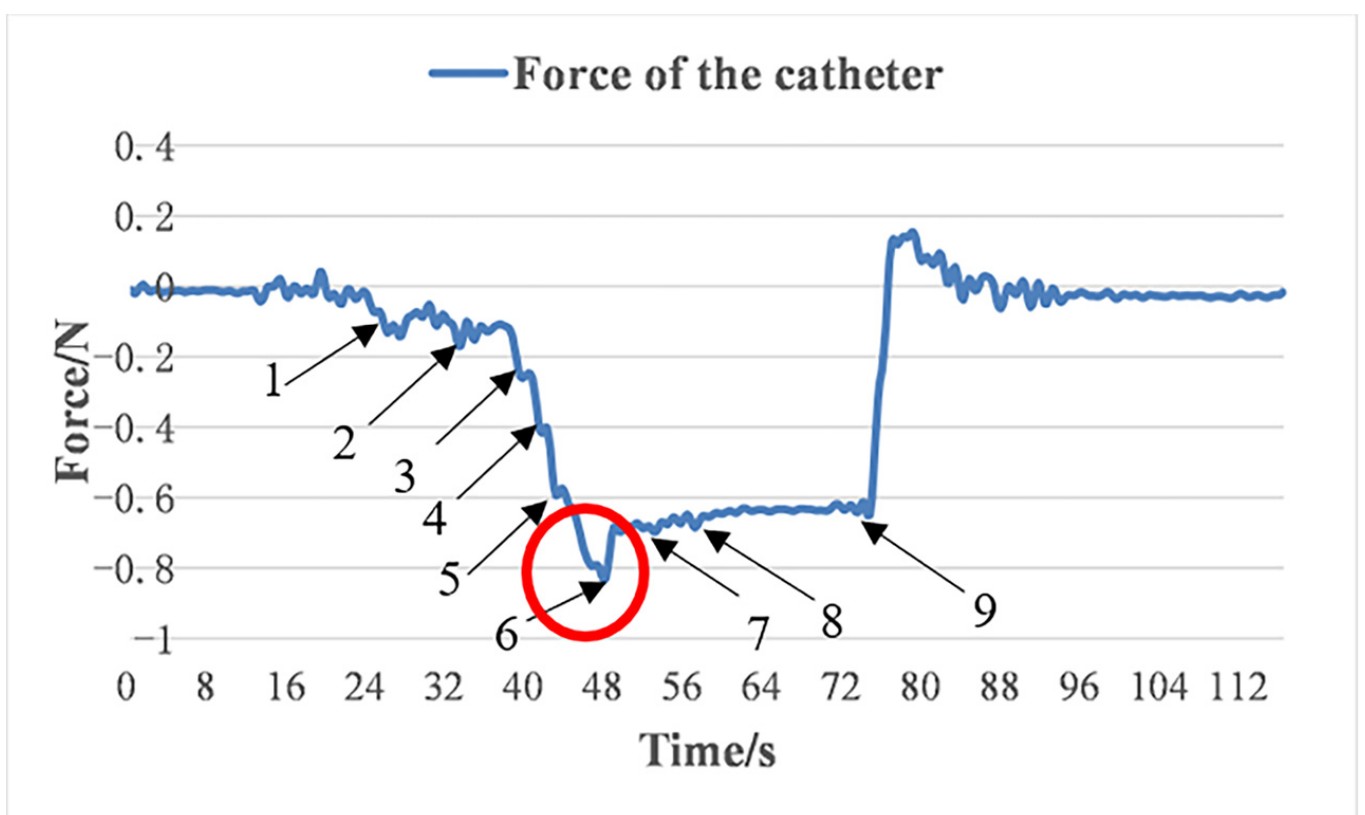

**Figure 17.** The force of the catheter under condition 1.

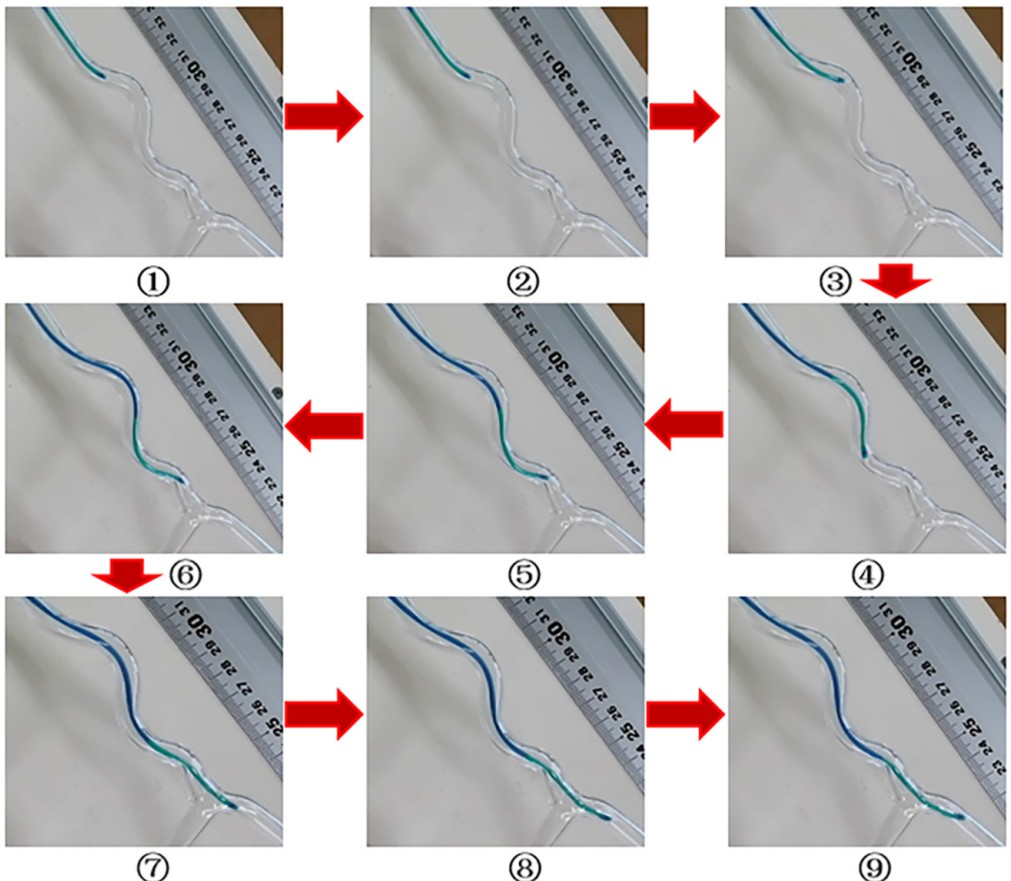

**Figure 18.** The corresponding position of the catheter in the glass tube as shown in Figure 17.

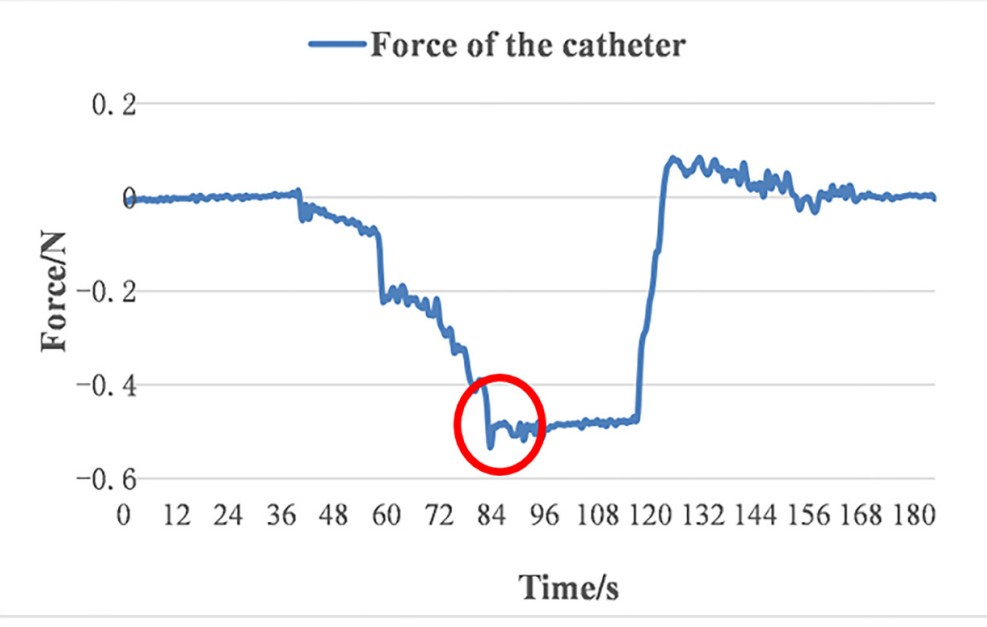

**Figure 19.** The force of the catheter under condition 2.

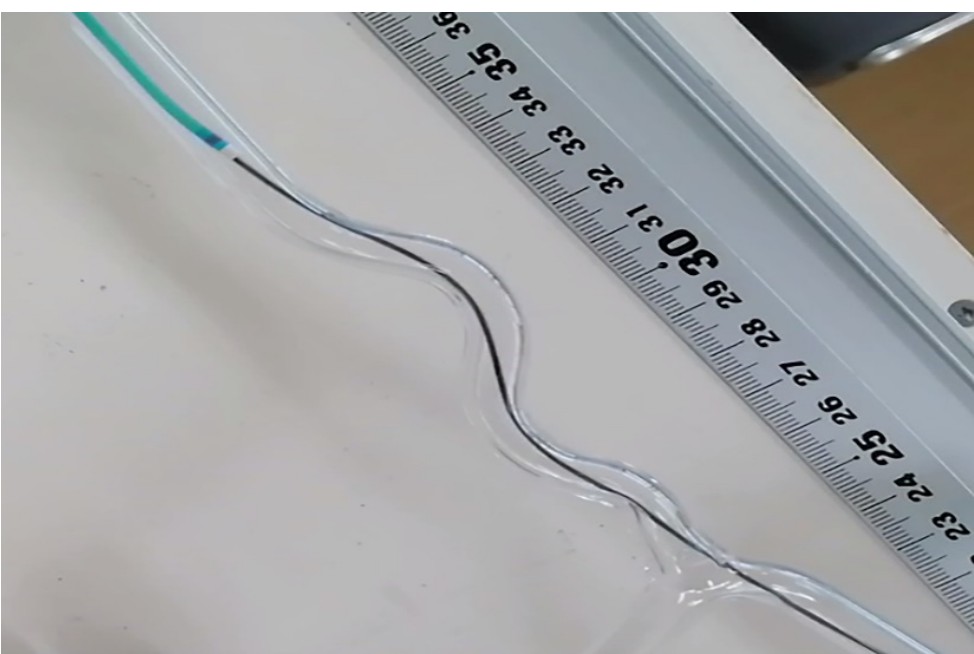

**Figure 20.** The initial situation under condition 2.

As shown by the red circles in Figures 17 and 18, compared with the case of condition 1, the force of the catheter under condition 2 did not change suddenly because this condition reduced the maximum number of CCE and prevented the tip of the catheter from coming into serious conflict with the inner wall of the glass tube. Under condition 1, the force ranges from 0 to 0.82 N. Under condition 2, the force ranges from 0 to 0.53 N. From the perspective of the catheter's force during the whole test process under condition 2, the force of the catheter is smaller than that under condition 1. Therefore, the guidance of the guidewire can effectively reduce the force of the catheter. We believe that in the complex and tortuous vascular path, planning the catheter's path will help to reduce the forces on the catheter, thereby reducing the risk of the blood vessel being punctured.

## 5. Discussions

In the force detection structure design, we used two torque sensors and a load cell to detect the radial force and axial force of the catheter/guidewire. It is worth mentioning that the radial force detection and identification in this study creatively use two torque sensors. Their primary function is to compare the same or different radial forces detected by the two torque values. Whether the catheter/guidewire has slipped during their rotation can be judged. The axial force calibration experiment results are different from previously reported studies [25,28]. These are mainly due to the flexible materials used in our slave system. Therefore, from the perspective of force measurement, flexible materials can effectively protect the catheter surface from damage [29], but they also affect the force measurement accuracy. The relationship between the accuracy of force measurement and the properties of flexible materials still requires further research.

The motion form of vascular intervention surgery is complicated. It includes advancing/retracting and rotational movement of the catheter/guidewire independently or simultaneously. In the glass tube test in this paper, the simple forward and backward motion of the catheter's axial force was recorded. The measured axial force was not affected by the rotational motion of the catheter. In practice, adding the rotational motion of the catheter makes the result a complex process. It is worth emphasizing that the load cell is installed behind the catheter. The axial force of the catheter measured by a load cell includes the resultant force of various forces acting on the axial direction of the catheter. The various forces mainly come from two aspects. For example, the contact force and

friction forces. The friction forces mainly include the catheter and the glass tube inner wall, the catheter, and the outer surface of the guidewire.

For future practical application in surgery, the size of the system will be miniaturized to facilitate surgeons better to operate master manipulation and slave manipulator access to the patient. In addition, the current form of clamping will be improved for better control of the catheter/guidewire's movement.

## 6. Conclusions

Robot-assisted endovascular intervention surgery has attracted significant attention in recent years. More and more types of robot-assisted endovascular intervention robots are in the process of being developed. These robots can be used in treating cardiovascular and cerebrovascular diseases. These diseases need to be treated in a timely manner. Thus, timely treatment is very crucial for patients. When the surgeon manipulates the robots, the robot's structural stability is always important. The robots can follow the traditional surgeons' operation habits to the maximum extent, which is beneficial to improve the transparency of an operation.

In this study, a master–slave robotic system was developed. It has force feedback, operates remotely, and follows the operating skills of traditional surgeons. To the best of our knowledge, there is generally a lack of axial and radial force detection functions in the current slave systems' structural design that can quickly clamp/release the catheter/guidewire. We are the first to design and manufacture this slave system structure. The slave system has good power transmission and structural stability. The advancement (retraction) and rotation of the catheter/guidewire can be operated simultaneously or independently. The test results show that the measurement accuracy of the axial force meets the requirements of robot-assisted surgery and the system can track the designed position of the catheter/guidewire in real time.

In the future, we will continue to improve the slave manipulator's prototype and use advanced controllers to improve the system's accuracy. In addition, the radial force measurement accuracy of the slave system will be verified through experiments.

**Author Contributions:** C.S. and S.G. conceived and designed the slave system, performed the data collection, and drafted sections of the manuscript. M.K. guided the evaluation experiments of the slave system. All authors have read and agreed to the published version of the manuscript.

**Funding:** This research is partly supported by the National High-tech Research and Development Program (863 Program) of China (No.2015AA043202), in part, by the SPS KAKENHI Grant Number 15K2120 and China Scholarship Council Grant Number 202208050086.

**Institutional Review Board Statement:** Not applicable.

**Informed Consent Statement:** Not applicable.

**Data Availability Statement:** The data presented in this study are available on request from the corresponding author. The data are not publicly available due to potential patent application.

**Conflicts of Interest:** The authors declare no conflict of interest.

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
