# Peer review of "Design and Performance Evaluation of a Novel Slave System for Endovascular Tele-Surgery"

_machines, doi:10.3390/machines10090795_

Round 1

Reviewer 1 Report

This paper proposes a slave system for endovascular tele-surgery, which has precise force feedback and can quickly clamp the catheter/guidewire. The slave system is compatible with various standard catheter/guidewire sizes and has a stable structure. The innovation of the paper is apparent, and the organization of the paper is logical. However, some minor issues in the paper need attention and modification.

Below are my detailed comments:

1.        The paper mentions the slave system can accurately feedback the axial force and radial force, but in the experiments, only the axial force feedback has been tested. It’s also necessary to test and show the accuracy of the radial force feedback.

2.        In Sec 3.3, the paper explains the reason for using two torque sensors is it can be judged whether the catheter/guidewire has slipped during the rotation process, which increases the safety of the operation. However, avoiding the catheter/guidewire slips is more important than detection, but this method of clamping a line between two planes is the fundamental reason for this problem.

3.        In Sec 4.1, the paper notes that “due to the influence of the flexible material coefficient, the relationship between F1 and F2 is about 1.33”. How do you get the coefficient of 1.33?

4.        In Fig 16(a), to complete the glass tube test, the slave system is extremely close to the glass tube. When it comes to endovascular telesurgery, the glass tube should be vascular. I think the slave system is too big to install so close to the vascular in the surgery.

5.        In Sec 3.2, the paper notes, “As shown in Figure 7, … the structure on the SP did not slide”. But in figure 7, the angle  is significantly smaller than, which makes the conclusion unconvincing.

6.        The figures in the paper are totally disordered; for example, in Sec 3.1, “the FP structure is shown in Figure 6”, which is actually in Figure 5. In Sec 4.1, “the left side of Figure 11 is a load cell fixed on the ball screw”, but figure 11 is an assembly figure with three sub-figures. Also, in Sec 4.1, “ is subtracted from … as shown in Fig 13”, but figure 13 shows the line tracking test setup on the slave side in Sec 4.2. 

Reviewer 2 Report

Presented paper demonstrates innovative approach for the development of the endovascular tele-surgery. Designed master-slave system can significantly improve surgery processes and minimize negative effects to the personal and patient. Authors give detailed information concerning system design and test applications. Paper written with good scientific language and contains all necessary information. I suppose paper require only minor revision before it can be accepted for publication. Main comments are:

1.      Figure 12, please make it a little bit bigger because in current conditions it’s difficult to read;

2.      In discussion text, please add brief overview of further development of system for future practical application in surgery;

3.      Line 170, please give more information concerning magnetorheological fluid that applied for the force feedback.

Reviewer 3 Report

Authors have presented a novel design for a master-slave device to be used in medical applications.

The paper is generally well written and easy to follow. However, I ask the authors to improve image quality: most of the images are blurred and sometimes difficult to read.  Moreover, in Figure 8 it is difficult to identify the components of the system (e.g., grasper A and slider A seem to refer to the same arrows, please use different colors).
